# Recent Progress of Natural Mineral Materials in Environmental Remediation

**Ningxin Kang [1], Weichuang Zhou [1], Zheng Qi [1], Yuhan Li [2,\*], Zhi Wang [1], Qin Li [1] and Kangle Lv [1,\*]**

[1] Key Laboratory of Resources Conversion and Pollution Control of the State Ethnic Affairs Commission, College of Resources and Environment, South-Central Minzu University, Wuhan 430074, China

[2] Chongqing Key Laboratory of Catalysis and New Environmental Materials, Chongqing Technology and Business University, Chongqing 400067, China

[\*] Correspondence: lyhctbu@126.com (Y.L.); lvkangle@mail.scuec.edu.cn (K.L.)

**Abstract:** Organic contaminants, volatile organic compounds (VOCs), and heavy metals have posed long-term threats to the ecosystem and human health. Natural minerals have aroused widespread interest in the field of environmental remediation due to their unique characteristics such as rich resources, environmentally benign, and excellent photoelectric properties. This review briefly introduced the contributions of natural minerals such as sulfide minerals, oxide minerals, and oxysalt minerals in pollution control, which include organic pollution degradation, sterilization, air purification (NO VOCs oxidation), and heavy metal treatment by means of photocatalysis, Fenton catalysis, persulfate activation, and adsorption process. At last, the future challenges of natural mineral materials in pollution control are also outlooked.

**Keywords:** natural mineral; photocatalysis; environmental remediation; advanced oxidation processes (AOPs)

## 1. Introduction

In today's world, the rapid growth of the world's population and industrial development has produced massive expansion of economic activities such as industry and agriculture, yielding contaminants harmful to shadow the ecological systems [1]. Anthropogenic activities give rise to significant imbalances of air, aqueous, and soil matrices, due to climate change and environmental pollution [2]. Seriously, it is fairly urgent to take some effective measures to adopt environmental remediation increasingly. Statistically, current primary pollutants include organic matter [3], volatile organic compounds (VOCs) [4], heavy metal ions [5], hazardous gases [6], and dyes [7]. Hence, efficient and economic environmental remediation technologies are supposed to be considered in order to contain and reduce these pollutants to safe concentrations.

Photocatalysis provides a sustainable way to solve environmental problems [8]. When light energy is irradiated onto the surface of a semiconductor photocatalyst, the photogenerated electrons can bind with surface adsorbed oxygen to produce superoxide radicals ($\bullet O_2^-$) [9]. Then, the surface of the photocatalyst becomes positively charged, which can take electrons from surface adsorbed water, resulting in the production of hydroxyl radicals ($\bullet OH$) [10]. Reactive oxygen species (ROS) such as $\bullet O_2^-$ and $\bullet OH$ radicals are powerful enough to attack organic compounds and turn them into water or other harmless substances.

$TiO_2$ is considered the star semiconductor photocatalytic material due to its strong oxidation ability, high photochemical stability, and excellent biocompatibility [11]. However, the pure anatase $TiO_2$ has a band gap of 3.2 eV, which therefore can only be excited by ultraviolet light with the utilization of less sunlight. Whereafter, it has been investigated that graphitic carbon nitride (g–$C_3N_4$) has a narrow band gap with a strong visible light response, which has aroused a strong response in the field of photocatalysis. However, it

has the disadvantages of low quantization yield and easy recombination of photogenerated electron holes [12]. Nevertheless, the efficiency of semiconductor photocatalysis is still difficult to meet the practical requirements at present and the high cost of catalyst preparation also be considered a challenge in the field of remediation.

Similar to synthetic semiconductor materials, some natural minerals have photocatalytic properties. Under light excitation, electron transitions occur in the valence bands of semiconductor minerals, resulting in the production of electron-hole pairs, which can initiate the photocatalytic oxidation of organic pollutants [13]. As is shown in Figure 1A, Professor M.F. Hochella from America revealed that the natural iron manganese ore widely distributed in earth exhibit highly responsive and stable photon-to-electron conversion similar to semiconductor materials. The element mapping from Figure 1B explained that this is due to surface weathering gradually resulting in the conversion of minerals to semiconducting types of iron oxides and manganese oxides [14]. And the photocurrent response of Fe-Mn coating from Figure 1C shows a strong signal, which suggested that natural mineral semiconductors deserved to be further studied.

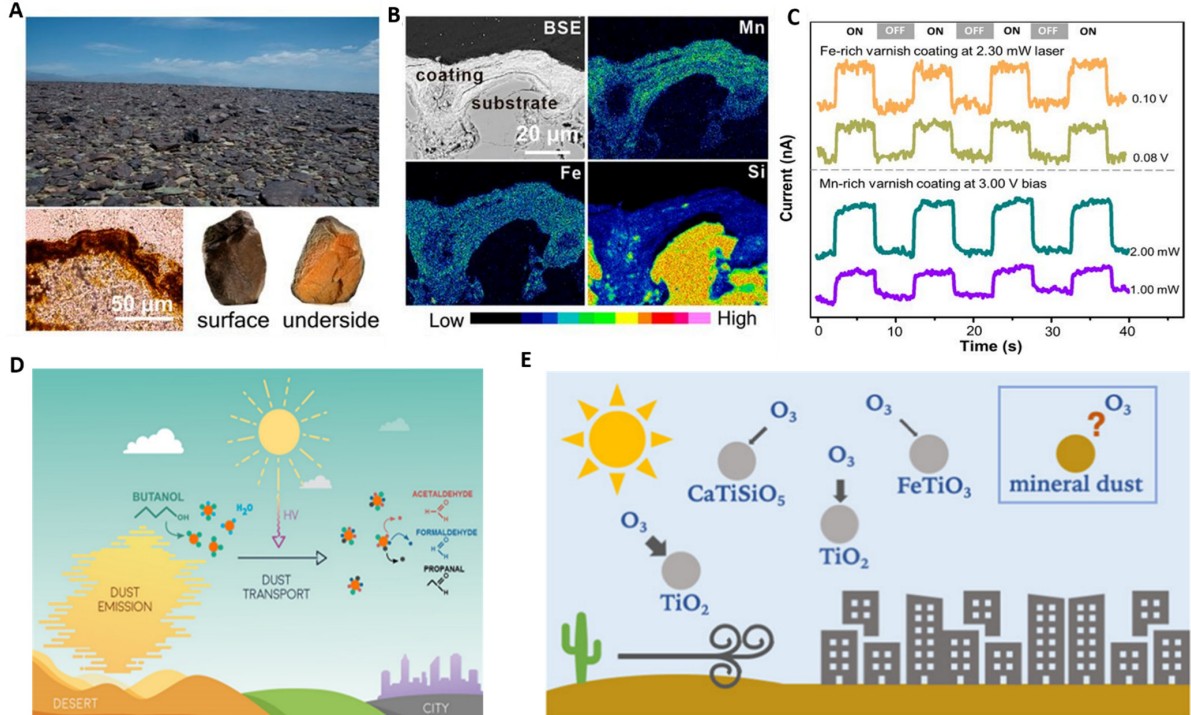

**Figure 1.** (**A–C**) Structure and catalytic properties of minerals: composition of ferromanganese ore and photoelectrical properties, copyright PNAS. (**D**) adsorption, migration, and photocatalytic oxidation of VOCs by mineral particles in air, copyright 2018 American Chemical Society. (**E**) photocatalytic decomposition of ozone in urban air by titanite dust, copyright 2020 American Chemical Society.

Professor C. George, a French scientist, studied the surface properties and photochemical behavior of natural mineral dust particles and clarified the chemical interactions between light and mineral [15], which revealed that natural mineral materials are expected to be used in air purification (Figure 1D). Similarly, Professor S.A. Styler, from the University of Alberta, Canada, investigated that a series of titanium minerals have a visible light catalytic decomposition of ozone in the air [16] (Figure 1E). Moreover, it has been studied that the heat-treated natural pyrrhotite (main components $Fe_2O_3$ and $FeS_2$) has excellent visible light sterilization performance [17] and the natural wolframite also has visible organic degradation and sterilization properties [18]. Xiaoyong Wu and co-workers disclosed that hematite nanocrystals exposed to different crystal planes affect persulfate (PS) activation for contaminant degradation under visible light irradiation [19]. These studies above have laid a solid foundation for the application of natural minerals in environmental remediation.

In fact, natural mineral materials are emerging hybrid materials for the treatment of pollutants in the field of environmental remediation. As the product of the natural evolution of the geological environment [20], natural minerals contribute greatly to the regulation and purification of the natural environment and offer valuable solutions to emerging environmental challenges according to their interaction and coordination with the hydrosphere, atmosphere, pedosphere, and biosphere. Generally, the different components of natural minerals give them different chemical properties. Some sulfide minerals have been proven to be conducive to visible-light photocatalysis, due to their narrow gap and excellent photoelectric property [21]. In addition, some minerals including transition metals or metal oxides usually cover the variable chemical states and empty orbitals, which can donate electrons to activate persulfate molecules [22]. Besides, some oxygenated salt minerals can be used as carriers or adsorbents, due to their high crystallinity [23], ultra-high surface area [24], and rich chemical functionality [25]. Meanwhile, serving as an important constituent of the earth's ecological environment, the properties of physics, chemistry, and biology about natural minerals are well coordinated with the ecological environment, which makes extensive progress using in the process of environmental remediation.

However, how to use natural mineral materials to improve the natural environment and carry out effective environmental self-purification lacks in-depth mechanism analysis and research summary. In this paper, we reviewed the relevant research progress of natural mineral materials in environmental remediation, including the classification of natural minerals and specific reaction processes of natural mineral materials in the field of environmental remediation. Meanwhile, the existing problems and the future development direction of natural mineral materials in the field of environmental remediation are also analyzed and summarized. It is fairly expected that this paper can provide reference information for the practical application of natural minerals in the field of environmental remediation.

## 2. Classification of Natural Mineral Materials

Natural mineral materials are natural elements or compounds formed under the comprehensive action of various materials in the Earth's crust. Meanwhile, the chemical composition and connection of natural minerals determine their unique physicochemical properties and thus have a decontamination function for some pollutants. On the base of the mineral element composition, it can be precisely divided into sulfide mineral, oxide and hydroxide mineral, oxysalt mineral, and other minerals.

### 2.1. Sulfide Minerals

In recent years, sulfides have become a hot topic due to their excellent catalytic activity, effective cost, and promising potential for activating peroxy-mono-sulfate to efficiently oxidize and degrade organic pollutants. Furthermore, the sulfides possess the advantages of great thermal conductivity and low carrier mobility, which is beneficial to use photocatalysis for environmental remediation [26]. For instance, chalcopyrite($CuFeS_2$) has the excellent property to activate peroxydisulfate (PDS) for the simultaneous degradation of organic pollutants [27]. Accordingly, natural chalcopyrite ($CuFeS_2$) makes it possible for in-situ remediation, immensely reducing the repair cost. [28]. In addition, it has been proven that pyrite ($FeS_2$) has remarkable performance in degrading organic matter [29], photocatalytic sterilization [17], and removing heavy metal ions [30].

### 2.2. Oxide Minerals

Oxides or hydroxide minerals are generally synthesized by calcination, and different phases of oxides may be formed by calcination, which are listed in Table 1. For instance, a micrometer-scale $ZnO/ZnFe_2O_4$ coupled photocatalyst was obtained by heat treatment of natural Fe-bearing sphalerite ((Zn, Fe) S). Among oxide minerals, hematite ($Fe_2O_3$), goethite ($\alpha$-FeOOH), and ilmenite ($FeTiO_3$) can be typically considered. For example, hematite nanocrystals have been considered in advanced oxidant processes (AOPs) due to their cost-effectiveness and eco-friendly, which are able to activate persulfate in order to achieve environmental reme-

diation [22]. Furthermore, it is reported that goethite ($\alpha$-FeOOH) has been utilized to construct the Fenton system, contributing to removing multiple organic contaminants effectively [3]. Moreover, plasma-treated goethite nanoparticles have the characteristic of high surface area and surface hydroxyl groups, enhancing the performance of ozone removal [31]. Besides, it has been investigated that ilmenite ($FeTiO_3$) has promising prospects in photocatalytic and catalytic processes, with a proper band gap varying at 2.4–2.9 eV [32].

**Table 1.** Summary of preparation method, reaction process, and practical application of different types of natural minerals in environmental remediation.

| Materials | Main Ponents | Reaction Process | Preparation Methods | Composition in Catalytic System | Light Source | Pollutants | Activity/Removal Efficiency | Ref. |
|---|---|---|---|---|---|---|---|---|
| Clinoptilolite | $SiO_2/Al_2O_3/CaO$ | Photocatalysis | Hydrothermal | Clinoptilolite/BiOCl/$TiO_2$ | Xe lamp | Sodium isopropyl xanthate | 90% (3 h) | [24] |
| Kaolinite | $Al_2O_3/SiO_2$ | Photocatalysis | Sol-gel | Kaolinite/$TiO_2$ | Xe lamp | Ciprofloxacin | $0.00597\ min^{-1}$ | [33] |
| Montmorillonite | $Al_2O_3/MgO/SiO_2$ | Photocatalysis | Thermal | Montmorillonite/$Bi_2O_3$/Ag | LED | Tetracycline | 90% (60 min) | [34] |
| Iron minerals | $FeS_2$ | Photocatalysis | Calcination | $FeS_2/SiO_2$ | UV-lamp | Bisphenol A | 99% (120 min) | [35] |
| Natural magnetite | $Fe_3O_4$ | Photocatalysis | Hydrothermal | Magnetite /$TiO_2$ | Xe lamp | Defotaxime | 52.5% (60 min) | [36] |
| Natural porous diatomite | $SiO_2$ | Photocatalysis | Calcination | $TiO_2$/diatomite | Hg lamp | Rhodamine B | 80% (60 min) | [37] |
| Natural wolframite | $FeWO_4/MnWO_4$ | Photocatalysis | Hydrothermal | - | Xe lamp | *E. coli* | 80% (4 h) | [18] |
| Natural Fe-bearing sphalerite | $ZnO/ZnFe_2O_4$ | Photocatalysis | Thermal | - | LED | *E. coli* | 100% (4 h) | [38] |
| Natural magnetic sphalerite | $ZnS$, $FeS_2$ | Photocatalysis | Ball milling | - | LED | *E. coli* | 100% (6 h) | [39] |
| Natural pyrrhotite | $Fe_2O_3$–$FeS_2$ | Photocatalysis | Thermal | - | LED | *E. coli* | 100% (4 h) | [40] |
| Natural magnetic sphalerite | $ZnS/ZnFe_2O_4$ | Photocatalysis | Calcination | $ZnS/ZnFe_2O_4/ZnO$ | LED | Gram-negative Escherichia coli K-12 | 100% (6 h) | [17] |
| Celestite | $SrSO_4$ | Photocatalysis | Calcination | Celestite/g-$C_3N_4$ | LED | NO | 67.5% (10 min) | [41] |
| Illite | $SiO_2/Al_2O_3$ | Photocatalysis | Calcination | Illite/g-$C_3N_4$ | Xe lamp | NO | 70% (6 min) | [42] |
| Attapulgite | $MgO/SiO_2$ | Photocatalysis | Impregnation | Attapulgite/$SmFeO_3$ | Xe lamp | NO | 90% (120 min) | [43] |
| Perovskite | $CaTiO_3$ | Photocatalysis | Sol-gel | Perovskite/N-CQDs | Xe lamp | NO | 75% (30 min) | [44] |
| Clay brick sands and recycled glass | $SiO_2/Al_2O_3/Fe_2O_3$, $CaO/MgO/Na_2O$ | Photocatalysis | Impregnation | Clay/$TiO_2$ | UV-lamp | NO | 75% (10 min) | [45] |
| Hematite | $Fe_2O_3$ | Persulfate oxidation | Calcination | - | - | Tetracycline | 70% (5 min) | [22] |
| Pyrite | $Fe_2O_3$ | Persulfate oxidation | Ultrasonic | - | - | 2,4-dichlorophenol Cr(VI) | 76.2% (120 min) 80.1% (120 min) | [46] |
| Goethite | $\alpha$-FeOOH | Fenton catalysis | Hydrothermal | - | - | Alachlor | 90% (6 min) | [3] |
| Natural wolframite | $FeWO_4/MnWO_4$ | Fenton catalysis | Hydrothermal | - | - | Methylene blue | 99% (3 h) | [18] |
| Hematite | $Fe_2O_3$ | Fenton catalysis | Solvothermal | - | - | Methylene blue | 80% (10 h) | [36] |
| Montmorillonite | $Al_2O_3/MgO/SiO_2$ | Adsorption | Heat treatment | Montmorillonite/biochar | - | Tetracycline | 77.962 mg/g | [47] |
| Montmorillonite | $Al_2O_3/MgO /SiO_2$ | Adsorption | Calcination | Ti-montmorillonite | - | Imipramine | 82.68 mg/g | [48] |
| Zeolites | $SiO_2/Al_2O_3$ | Adsorption | Grinding | - | - | $Ni^{2+}$ | 105.93 mg/g | [49] |
| Bentonite | $SiO_2/Al_2O_3/Na_2O/CaO$ | Adsorption | Grinding | - - | - - | dichloromethane Hg(II) | 119.93 mg/g 21.29 mg/g | [50] [51] |
| Attapulgite | $SiO_2/MgO$ | Adsorption | Grinding | - | - | Cd(II) | 117.8 mg/g | |
| organo-Montmorillonite | $Al_2O_3/MgO/SiO_2$ | Adsorption | Grinding | - | - | Pb(II) | 1.0803 mmol/g | [52] |
| Magnetic geopolymer | $SiO_2/Al_2O_3$ | Adsorption | Grinding | - | - | Cu(II) Ni(II) Cd(II) | 440 mg/g 400 mg/g 380 mg/g | [53] |

### 2.3. Oxysalt Minerals

2.3.1. Silicate Minerals

Silicate minerals are a kind of oxygen-bearing acid minerals formed by the combination of metal cation and silicate, estimated to account for more than 90% of the entire crust, including layered clay minerals, lamellar chain structure of palygorskite (Si/SiOx) and sepiolite ($Si_{12}Mg_8O_{30}(OH)_4(OH_2)_4 \cdot H_2O$), and minerals with shelf, chain, island, and other structures. Moreover, the methods of synthesis of silicate minerals include the hydrothermal method, sol-gel method, water bath precipitation method, and so on, which are listed in Table 1. Statistically, both palygorskite (Si/SiOx) and sepiolite belong to porous clay minerals with a high specific surface area. Palygorskite (Si/SiOx) has the unique characteristic of rod-like structure, large surface area, and tunable chemistry, which attracted massive interest in environmental remediation [54]. The large surface area of palygorskite (Si/SiOx) enables organic matter to reach the active center, which is extremely beneficial for removing contaminants from water. Similarly, sepiolite ($Si_{12}Mg_8O_{30}(OH)_4(OH_2)_48H_2O$) is a mineral with molecular-sized pores. It is described as oxygen-oxygen and hydroxide bound to magnesium ion centers, due to the molecular organization of a unit cell stacked in a 2:1 ratio. Because there are discontinuous octahedral lamellae in the sepiolite structure, some porous channels enable pollutants to enter the surface structure rapidly. Therefore, it has a promising potential in environmental remediation such as degradation of organic matter and adsorption of heavy metal ions [23,34–36].

Kaolinite ($Al_2O_3/SiO_2$), as a low-cost, easily available, and pollution-free material, has been gradually applied in the field of environmental water treatment. Kaolin ($Al_2O_3/SiO_2$) is a 1:1 layered aluminosilicate clay mineral formed by a layer of tetrahedral silicon-oxygen and a layer of octahedral aluminum oxygen connected by O atoms, which is abundant in the Earth's crust. Each unit layer consists of a silicon-oxygen tetrahedral sheet [$SiO_4$] and an aluminum oxygen octahedral sheet [$AlO_6$], with adjacent layers connected by hydrogen bonds. The unique porous structure is favorable for the catalyst to disperse on the surface and has an excellent support function, which is conducive to removing heavy metal ions in soil and degrading pollutants in water remediation [17,28–31].

In addition, the crystal structure of zeolite is composed of silicon (aluminum)-oxygen tetrahedra connected into a three-dimensional lattice with holes and channels of various sizes [55]. The structural formula of zeolite is $A_{(x/q)}[(AlO_2)_x(SiO_2)_y] \cdot n \, (H_2O)$: where A is Ca, Na, K, Ba, Sr, and other cations, B is Al and Si, P is the cation valence, M is the number of cations, n is the number of water molecules, x is the number of Al atoms, y is the number of Si atoms, (y/x) is usually between 1 and 5, (x+y) is the number of tetrahedra in A unit cell. Therefore, cavities of different sizes present in the lattice can absorb molecules of other substances of different sizes [56]. Studies have shown that zeolite, as an inorganic porous material with high surface area and special microporous channels, is considered a promising adsorbent for VOCs control [57,58].

2.3.2. Borate Minerals

Borate minerals are considered a compound of a metal cation and borate radical. Typically, tourmaline, due to its unique photochemical properties, has been extensively studied in the field of environmental remediation. It has been investigated that tourmaline is a natural borosilicate mineral consisting of a complex structure. It is a silicate mineral with a ring structure characterized by aluminum, sodium, iron, magnesium, and lithium-containing boron. The special structure of tourmaline has the ability to release negative ions or generate an electric field on its surface, which makes it possible to apply in environmental remediation [59].

### 2.4. Other Minerals

Interestingly, it has been strongly verified that other minerals used to purify the environment are hydrotalcite, apatite ($Ca_5(PO_4)_3$), and rutile ($TiO_2$). Hydrotalcite is an anionic layered compound. Further, the typical hydrotalcite compound is Mg-Al carbonate

type hydrotalcite: $Mg_6Al_2(OH)_{16}CO_3 \cdot 4H_2O$. As a special material with great potential, hydrotalcite has unique properties such as high porosity, high specific surface area, excellent photostability, low cost, and adjustable band gap [60–66]. Hence, it can be fairly expected to define hydrotalcite as a carrier to improve photodegradation by concentrating organic contaminants on the catalyst's surface via adsorption [67]. In addition, hydroxyapatite is a potential material for carbon dioxide capture [68]. Natural rutile ($TiO_2$) is cheaper and easier to prepare than synthetic photocatalysts, which has been studied for dye removal due to its narrow band gap and easy formation of hydroxyl groups on the surface [69].

## 3. Applications Minerals in Environmental Remediation

Benefiting from multiple synergetic effects, the natural mineral materials exhibit excellent photoactivity, which opens new eyes for achieving the desired applications of catalysis in environmental remediation, including photocatalytic antibacterial, NO oxidization, oxidation/degradation of organic pollutants, reduction of toxic high-valence metal ions, and adsorption. Investigations on natural minerals endowed with environmental remediation are in the developing stages, with both fantastic opportunities and challenges coexisting. As discussed above, the unique porous structure and surface chemistry of natural minerals show great potential in environmental remediation. Here, combining the practical needs of environmental remediations and the advantageous properties of natural minerals are discussed above. This paper highlights several notable findings about natural minerals in the primary fields of environmental remediation, including water remediation, air purification, and soil restoration. Meanwhile, we find that natural minerals mainly play a role in environmental remediation through photocatalysis, Fenton catalysis, and adsorption reaction.

### 3.1. Sterilization and NO Removal by Photocatalysis

#### 3.1.1. Sterilization

Photocatalytic technology is an effective and sustainable method, which has attracted more and more attention as green disinfection. In this regard, natural minerals are considered to be the most promising photocatalyst due to the unique advantages of easy access to raw materials and low cost, and at the same time have great potential in practical applications. The bactericidal ability of natural minerals mainly depends on the photocatalytic reaction. The metal oxides contained in minerals are semiconductors and thus have photocatalytic capabilities. Typically, a naturally occurring ilmenite ($FeTiO_3$) was successfully synthesized to serve as an efficient photocatalyst. The result suggested that it had bacteriostatic properties with 6 log10 CFU/mL *E. coli* under visible light irradiation for 30 min. As is shown in Figure 2A, the high bactericidal effect was attributed to $\equiv$Fe(II) of ilmenite promoted persulfate activation vice and the accelerated capture of photo-generated electrons by persulfate versa to collectively generate more radicals for *E. coli* inactivation in the ilmenite/persulfate/visible light process and the improved photocatalytic activity for visible light owing to natural minerals enriched in transition metals activating the persulfate system [32]. By the combined utilization of persulfate (PS) and visible light (Vis) irradiation, the ilmenite/PS/Vis process is demonstrated by the result that the bactericidal efficiency increased fourfold than pristine (Figure 2B). In addition, a novel magnetic natural pyrrhotite ($Fe_2O_3$–$FeS_2$) mineral photocatalyst modified through thermal treatment was found to exhibit a remarkably enhanced bactericidal activity. As is observed in Figure 2C–E, the SEM showed that the bacterial cells are more vulnerable to powerful ROS [40]. Similarly, it is demonstrated that natural wolframite with a narrow band gap can generate electrons and holes under visible light to get involved in inactivating *E. coli* K-12 [18]. Besides, micrometer-scale $ZnO/ZnFe_2O_4$ coupled photocatalyst also exhibited excellent 100% lethality of *E. coli* under visible light irradiation [38]. These findings indicate that natural mineral materials have strong bactericidal potential, and the bactericidal potential is mainly attributed to the visible light response of transition metal oxides contained in natural minerals.

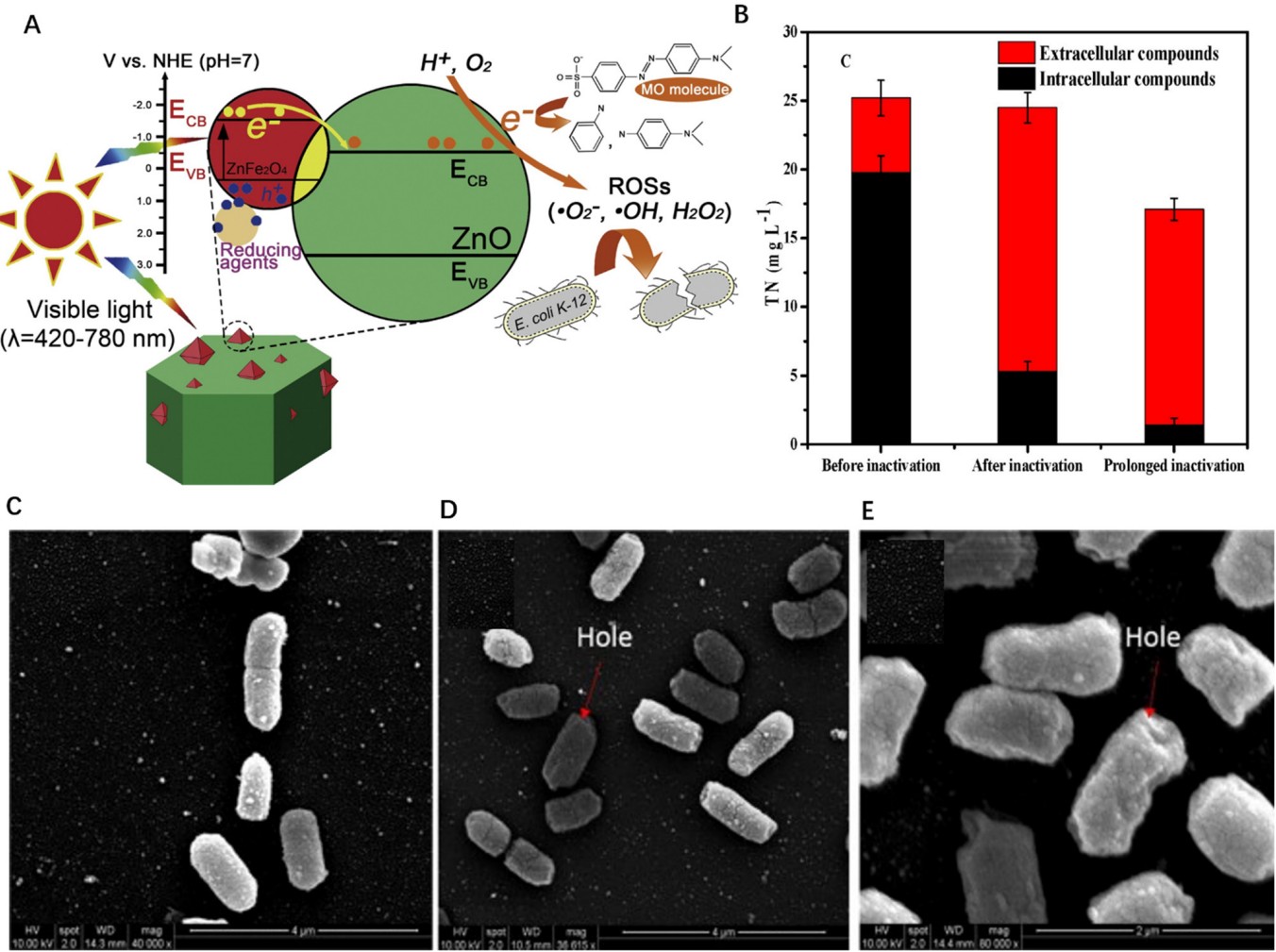

**Figure 2.** (**A**) Schematic illustration of photocatalytic sterilization mechanism of fabricated ZnO/ZnFe$_2$O$_4$., copyright 2017 Elsevier B.V. (**B**) Changes in total organic carbon before and after long-term inactivation of *Escherichia coli*, copyright 2018 Elsevier B.V. (**C–E**) SEM images of *E. coli* K-12 with photocatalytic inactivation treatment, copyright 2015 Elsevier B.V.

### 3.1.2. Nitrogen Oxide Removal

NO is one of the main pollutants that not only trigger disease but also cause great damage to the natural environment and our daily life. However, traditional de-nitration processes have high requirements for reaction conditions, such as selective catalytic reduction, which limits the application of these methods in real-life situations with low NO concentration [70]. Therefore, removing NO via photocatalytic material is of great significance [71]. Natural mineral, as a low-cost and available material, has been expected to be applied to NO removal. It has been proved that natural mineral materials can degrade nitrogen oxides through the photocatalysis process.

Interestingly, as is shown in Figure 3A, Qingxin Ma and co-workers found that mineral dust has a significant influence on the nitrogen oxides in the atmosphere. The result suggested that it is worthy of further study between mineral dust and the NO$_X$ transformation [72]. Hence, it is expected to study the effect of natural mineral materials on photocatalytic removal of NO, so as to design high-quality mineral photocatalysts for NO removal. The Sr ion in celestite (SrSO$_4$) contains empty orbitals, which help the photocatalyst to absorb more oxygen by coordination bonds [41]. Hence, the combination of celestite (SrSO$_4$) and photocatalyst may produce more reactive oxygen species. Combining illite particles (SiO$_2$, Al$_2$O$_3$) with photocatalysis has a similar property. The Si and AI of illite particles (SiO$_2$, Al$_2$O$_3$) are able to bridge the N in g-C$_3$N$_4$ to form the empty 3p orbital with

the coordination bond [73]. The result showed that the removal rate of modified material ($0.18 \, \text{min}^{-1}$) was 1.8 and 3.0 times higher than the pristine. In addition, perovskite ($CaTiO_3$), with the appropriate band structure, has gained increasing attention in photocatalysis. It must be stressed in Figure 3B that $CaTiO_3$ can serve as support to provide more active sites for the N-CQDs, which is beneficial for exciting electron and hole pairs [74]. The result showed that the removal of gaseous NO removal and the selectivity of $NO_2$ have been demonstrated much better than before (Figure 3C). Moreover, some silicate minerals can also act as support, which can assist the photocatalyst to remove NO. For instance, the addition of attapulgite ($Mg_5Si_8O_{20}(OH)_2(OH_2)_4 \cdot 4H_2O$) can not only uniformly disperse the composite catalyst to activate active sites but also absorb the NOx to react with the active sites in order to produce the final products of $NO_3^-$. The result showed that the utilization of natural minerals in air purification is helpful for catalysis to produce some active species with strong oxidation capability, but also absorb the target to react with those active species.

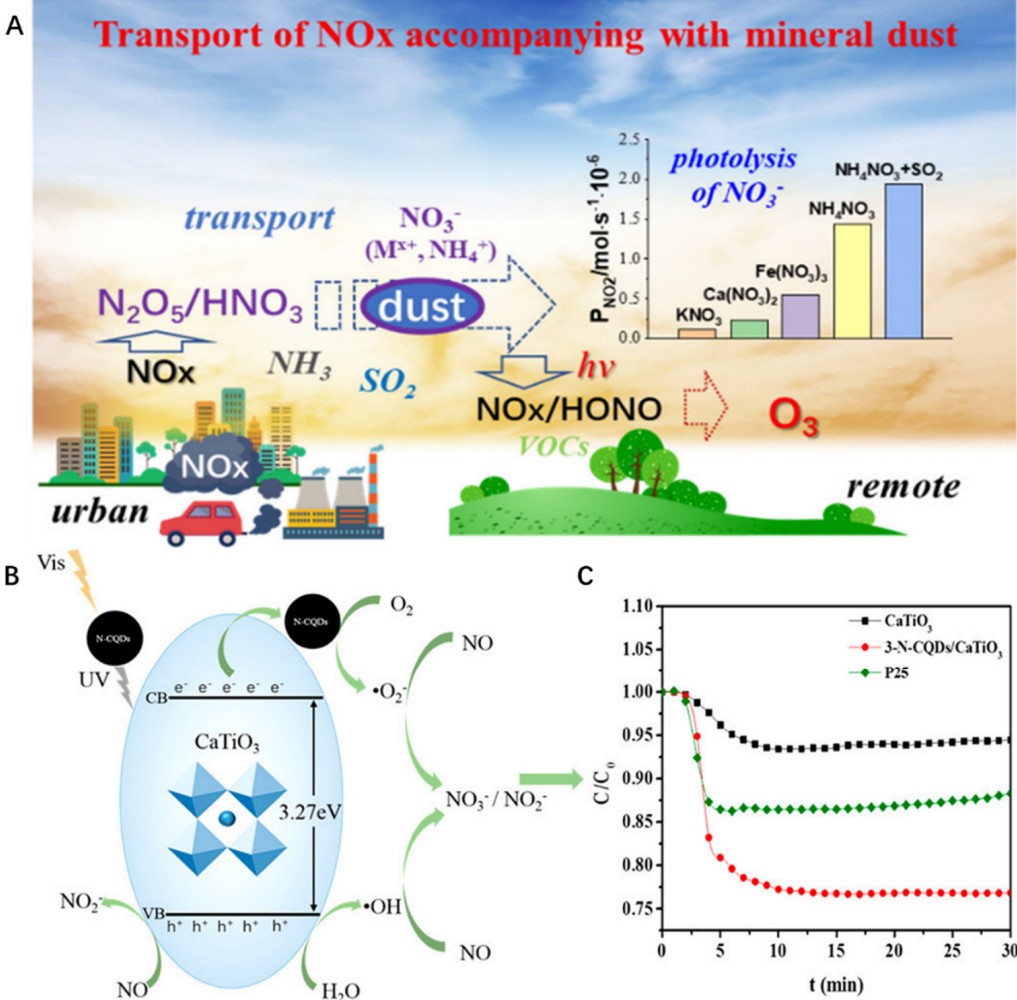

**Figure 3.** (**A**) The process of photooxidation of nitrates by minerals, copyright 2021 American Chemical Society. (**B**) Mechanism diagram of photocatalytic oxidation of nitrogen oxides of 3−N−CQDs/CaTiO₃ composite, copyright 2018 American Chemical Society. (**C**) Removal efficiency of nitrogen oxides by mineral photocatalytic oxidation, copyright 2018 American Chemical Society.

### 3.2. Water Purification by Advanced Oxidation Process

Advanced oxidation processes (AOP) such as the Fenton reaction and persulfate activation have been widely used in environmental remediation such as organic pollutants degradation due to the production of ROS such as $\bullet OH$ and $\bullet SO_4^-$.

It has been proved that the Fenton reaction can produce $\bullet OH$ reactive oxygen species in aqueous solution to degrade pollutants. Some scholars have studied the performance and mechanism of the removal of pollutants from water by the Fenton process, which uses natural minerals as catalysts. $Fe^{2+}$ plays a significant role in the Fenton reaction process, and the electron transfer of $Fe^{2+}/Fe^{3+}$ can catalyze the decomposition of hydrogen peroxide to produce $\bullet OH$ [75]. Therefore, it is expected to consider using the Fenton reaction of natural minerals to remove pollutants in water remediation, such as organic matter and heavy metals. In addition, some clay mineral has been proved to assist Fenton catalyst to degrade organic matter [76]. Therefore, the utilization of clay or iron oxide minerals as catalysts for the Fenton reaction is a promising alternative with great potential for environmental remediation [77].

#### 3.2.1. Degradation of Organic Pollutants by Fenton Catalysis

It has been observed that natural minerals have the ability to degrade organic matter, which is expected to play a great role in environmental water remediation. In the Fenton system with goethite ($\alpha$-FeOOH) as an iron source, a large number of hydroxyl radicals were generated, resulting in the degradation of bisphenol A [78]. The main principle of natural mineral removal of organic pollutants in water is to use an advanced oxidation process. Some natural minerals containing iron can activate hydrogen peroxide to produce Fenton or Fenton-like reactions. As is shown in Figure 4A, with the addition of $NH_2OH$, goethite can form a surface Fenton reaction and activate $H_2O_2$ to produce more $\bullet OH$ [3]. The result showed this surface Fenton system is able to degrade various organic pollutants, which has great practical application value in mineral remediation of the environment. Some scholars have compared the removal effect of imidacloprid by different natural iron minerals and zero-valent metals as Fenton catalysts, such as pyrite ($FeS_2$), ilmenite ($FeTiO_3$), vanadium titano-magnetite, zero-valent iron, and zero-valent copper [79]. The result revealed that the natural pyrite showed a high removal rate and stability, which was considered to be a promising Fenton catalyst. Similarly, it has been observed that natural pyrite can activate $H_2O_2$ to produce heterogeneous Fenton reactions to degrade tetracycline [29]. As is shown in Figure 4B, iron divalent is formed on the surface of pyrite and the main reactive oxygen species are $\bullet OH$ in the pyrite/$H_2O_2$ system. Moreover, the oxidation efficiency of tetracycline exceeds 85%. In addition, some metal oxides in natural minerals can activate persulfate to produce more reactive molecules to degrade pollutants. Some scholars synthesized a new type of pyrite nanosheet by the hydrothermal method. This pyrite nanosheet is oriented to form clusters of hexapo-nanosheets, which show an excellent ability to adsorb and degrade ciprofloxacin [80]. The result from Figure 4C reveals that the efficient Fe(III)/Fe(II) transformation has greatly promoted the activation of persulfate, simultaneous releasing more $SO_4^{\bullet-}$ and $\bullet OH$. Therefore, it can be concluded that the oxidation of refractory pollutants is mainly catalyzed by Fe(II) on the mineral surface. Surface Fe(II) reacts with $H_2O_2$ to produce more reactive oxygen species such as hydroxyl radicals [81]. Besides, some clay minerals can be used as support to provide more active sites for the catalyst. It has been observed that clinoptilolite ($SiO_2$, $Al_2O_3$, $CaO$), montmorillonite ($Al_2O_3$, $MgO$, $SiO_2$), and kaolinite ($Al_2O_3$, $SiO_2$) all have the ability to increase specific surface area, enhancing the performance to remove organic matters in water remediation.

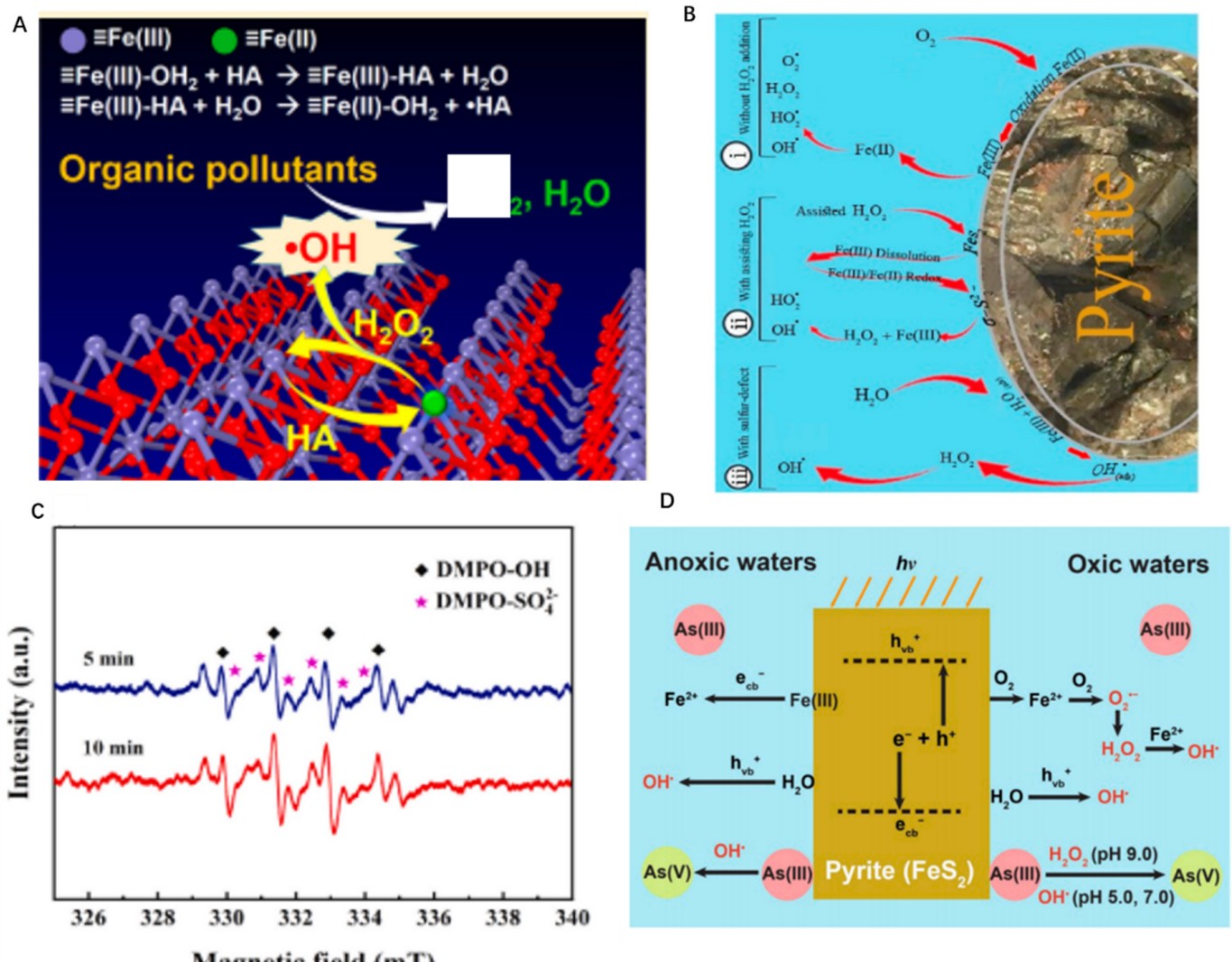

**Figure 4.** (**A**) Schematic diagram of degradation of organic matter by hydrogen peroxide activated by goethite, copyright 2017 American Chemical Society. (**B**) Schematic diagram of pyrite activation of Fenton reaction, copyright 2020 Elsevier Ltd. (**C**) Schematic diagram of free radical generation from persulfate system activated by pyrite, copyright 2020 Elsevier B.V. (**D**) The oxidation of As(III) by pyrite. Copyright 2021 Elsevier Ltd.

### 3.2.2. Oxidation of As(III) by Fenton Catalysis

Arsenic contamination in natural water has posed a great threat to millions of people in many regions of the world. Studies have found that some natural minerals can release iron minerals that can be used to oxidize As(III) in water by Fenton catalytic reaction. For instance, it has been observed that some sulfide minerals such as $CuFeS_2$ and $FeS_2$ have the capability of activating $H_2O_2$ [27]. As is shown in Figure 4D, the oxidation of $Fe^{2+}$ released from pyrite contributed much to the generation of $\bullet OH$, $\bullet O_2^-$, which is beneficial to remove As(III). The reactive oxygen species (ROS) effectively promote As(III) oxidation and adsorption on the pyrite surface [30]. Above all, natural minerals are a promising material to participate in the redox reaction to reduce the toxicity of the metals or nonmetals in wastewater.

### 3.2.3. Water Purification by Persulfate Activation

Some natural minerals are abundant in transition metals (oxides) such as iron, manganese, and copper, which are generally used for the chemical activation of persulfate (PS) to produce powerful ROS of $\bullet SO_4^-$ [3]. Removal of pollutants by persulfate oxidation

is another AOP. The use of metals to activate persulfate is a common method with the property of low energy consumption and high efficiency.

Nanoscale zerovalent iron, iron oxides, and pyrite ($FeS_2$), are promising candidates for activating PS/PMS in environmental remediation [29]. Xiangwei Zhang and co-workers first utilized natural illite($SiO_2$, $AI_2O_3$) micro-sheets to activate peroxy-mono-sulfate (PMS) [82]. These elements and their oxides can donate electrons to activate PS molecules due to their variable chemical states and empty orbitals. As is shown in Figure 5A, Tianming Cai and co-workers investigated that the addition of Fe mineral could effectively accelerate the redox cycle of Fe(III) $\leftrightarrow$ Fe(II) to produce ROS, contributing to improving the activation of peroxydisulfate(PDS) [83]. In addition, some physical approaches (radiation, heat, and sonication) also can be adopted to promote the degradation of organic pollutants via direct PS activation or facilitating Fe(III)/Fe(II) conversion. For instance, Fe(II) in ilmenite promotes the activation of PS and generates more free radicals to inactivate *Escherichia coli* under visible light irradiation [32]. Besides, as is shown in Figure 5B, natural chalcopyrite ($CuFeS_2$, NCP) has been investigated to activate peroxisulfate (PDS), which is able to simultaneously degrade the organic pollutants Rhodamine B (RhB) and reduce hexavalent chromium (Cr (VI)) [27]. Similarly, natural chalcopyrite is considered for use in wastewater remediation. It is observed in Figure 5C that the addition of natural chalcopyrite significantly activated the PDS system and produce massive free radicals to improve the degradation efficiency [28].

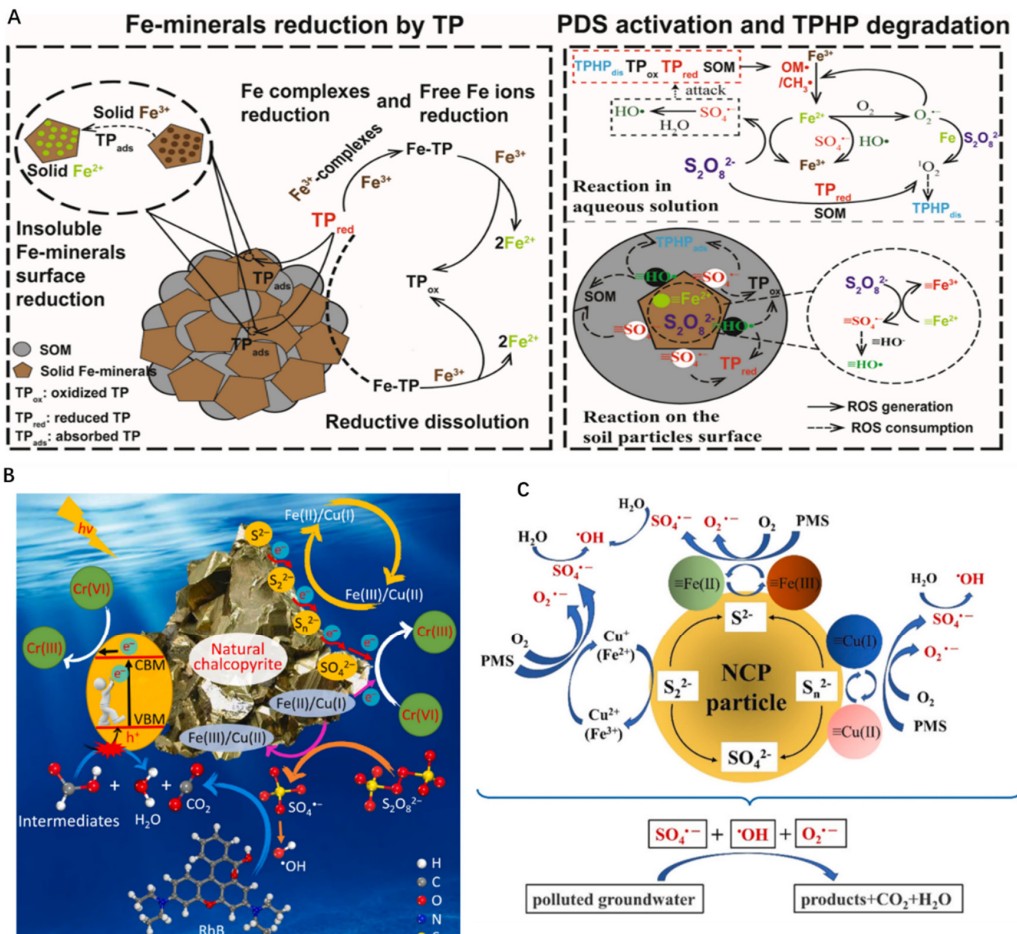

**Figure 5.** (**A**) The process of Fe-minerals reduction and the generation of ROS, copyright 2022 Elsevier B.V. (**B**) The mechanism of free radical production from persulfate system activated by natural minerals, copyright 2021 Elsevier B.V. (**C**) Schematic diagram for the activation mechanism of PMS by natural chalcopyrite, copyright 2021 Elsevier B.V. All.

### 3.3. Removal of Heavy Metals and Volatile Organic Compounds (VOCs) by Adsorption

Some clay minerals have bidimensional structures and abundant surface active groups, which can be defined as alternative adsorbents [84]. Therefore, it can be considered to use natural minerals for environmental remediation, such as soil restoration and air purification.

### 3.3.1. Adsorption of Heavy Metals

Studies have shown that natural minerals have been proven for soil remediation. For instance, zeolite ($A_{(x/q)}[(AlO_2)_x(SiO_2)_y]\cdot n(H_2O)$), with the microporous aluminosilicate frameworks, has the ability of ion exchange to absorb containments. It can be clearly observed in Figure 6A that the ion-exchange capacity and hydrophilicity of Si/Al can be significantly affected by adjusting Si/Al from 1 to infinity [50]. On this basis, Cheng-yu Chen and co-workers combined nano ferric oxide with zeolite to adsorb cadmium from wastewater [85]. In addition, adjusting the interlayer space and the porosity of the clay minerals through heating treatment can be adopted as an effective measure to enhance its absorption. Kexin Guo and co-workers used a natural clay palygorskite (Si/SiOx) to serve as a highly regenerable and efficient phosphate scavenger. The result disclosed that the co-calcination palygorskite (Pal) and La-based nanoparticles lead to the formation of $Al_2O_3$, $Fe_2O_3$, and MgO nanoparticles acting as new sites for phosphate adsorption, exhibiting excellent regeneration performance with high removal capacity [54]. As is shown in Figure 6B, it has been found that some clay minerals have the ability to adsorb heavy metals in soil. This is due to the excellent adsorption capacity of silicate clay minerals with a large specific surface area and porous structure. In conclusion, it is worth mentioning that the larger specific surface area and pore volume of natural minerals can provide more reaction sites, so the composites have better adsorption activity than pristine. Furthermore, synthetic minerals are expected to be promising functional adsorbents in environmental remediation. Some scholars have turned waste into wealth. They utilize some "waste", such as concrete and slag, to synthesize novel self-assembly adsorption materials (Figure 6C) [25].

Besides, it must be emphasized that the utilization of natural minerals is able to reach in situ immobilization, is non-disruptive, and cost-effective. For instance, natural sepiolite ($Si_{12}Mg_8O_{30}(OH)_4(OH_2)_48H_2O$) has a significant immobilization effect for heavy metals in soil. In addition, it has been observed that the hierarchical pore architecture of some clay minerals can selectively remove heavy metal ions in soil [50]. The discussion above proves that natural minerals can be used in soil remediation. Furthermore, Chunjie Yan and co-workers first studied that moderate thermal activation can improve the adsorption performance of sepiolite [86–88]. The result disclosed that thermal activation can tune the surface charge of sepiolite and generate more adsorption active sites. This finding can provide a new reference for the modification of other natural minerals.

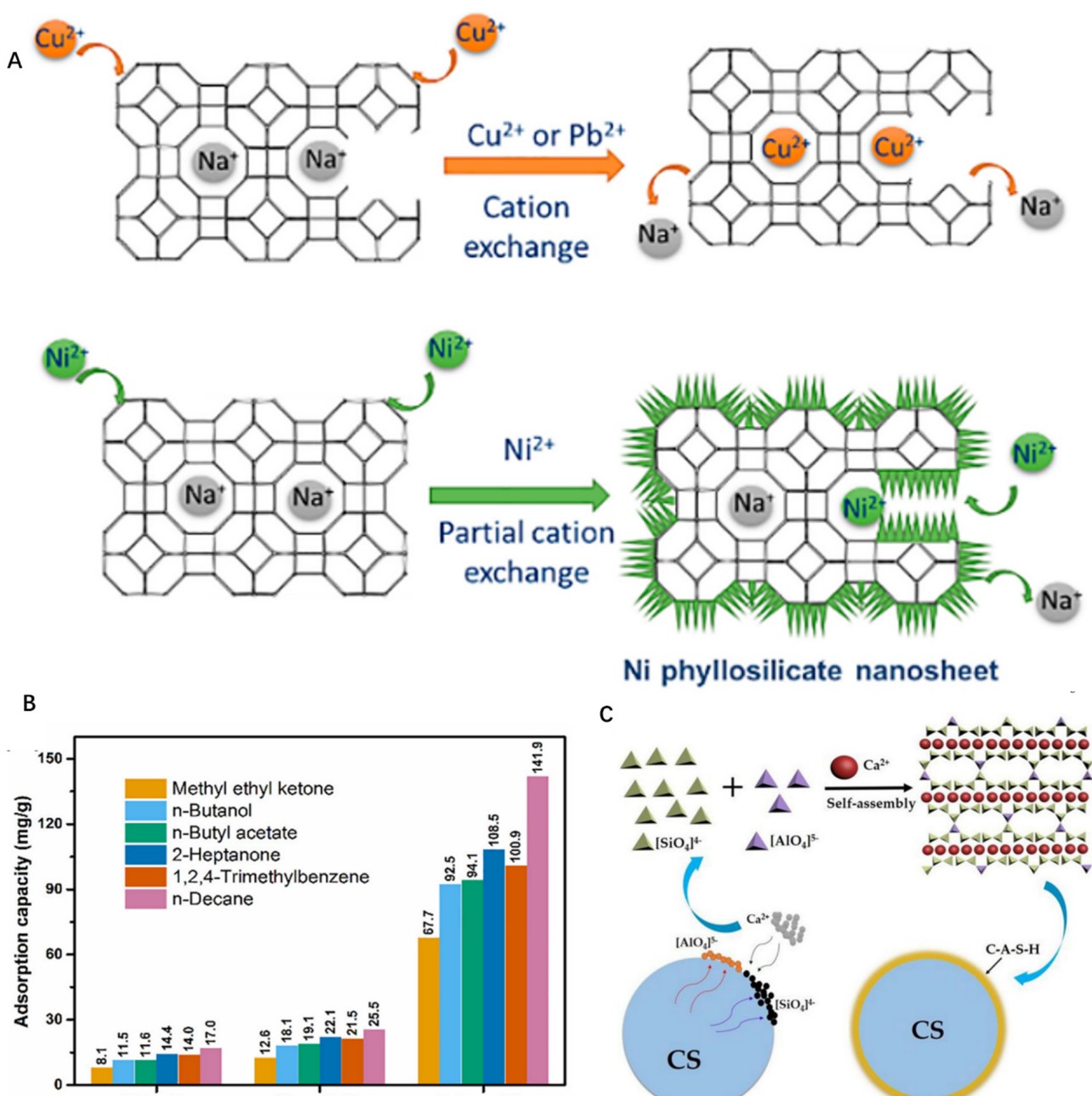

**Figure 6.** (**A**) Adsorption of heavy metals by zeolite, copyright 2018 Elsevier B.V. (**B**) Adsorption capacities of the diatomite, clinoptilolite, and palygorskite for VOCs, copyright 2019 Elsevier B.V. (**C**) Synthetic artificial self-assembly minerals, copyright 2018 Elsevier B.V.

### 3.3.2. Volatile Organic Compounds (VOCs) Adsorption

VOCs pollutants are carbon-containing organic chemicals from a wide range of sources, including outdoor sources (such as chemical production, automobile exhaust, and so on). Studies show that natural minerals could be attractive candidate adsorbents [4]. It has been proven that adjusting the proportion of oxides to control the morphology of minerals can be regarded as an effective means to improve mineral properties. For instance, some researchers synthesized zeolite with adjustable morphology and structure, which showed good adsorption performance of VOCs. On the other hand, it is also an excellent modification strategy to combine natural minerals with other mesoporous materials to construct fresh structures. The research revealed that the optimal composite of zeolite and silica showed significant VOCs adsorption performance.

## 4. Conclusions

Natural minerals are widely distributed on earth, and some of them have stable chemical structures with excellent electrochemical performance, which therefore are promising candidates used for environmental remediation. This review summarized the applications of natural minerals in water and air purification such as sterilization, NO and VOCs oxidation, and heavy metals removal using the techniques of photocatalysis, advanced oxidation (Fenton catalysis and persulfate activation), and adsorption.

Natural mineral materials have unique advantages of abundant resources, low price, and environmentally benign. However, they suffer from low efficiency from the viewpoint of practical applications. In addition, the complex structure of natural mineral materials makes it difficult to determine the active component. In future research, the following problems related to natural mineral materials deserve our attention:

(1) Some mineral materials have a low content of transition metal compounds. In order to achieve the same catalytic effect as the artificial metal catalyst, the natural mineral should be modified. For example, impregnation and roasting are considered to further improve the pore structure of natural mineral materials and increase the active sites on the surface of catalysts, so as to strengthen the activity of mineral materials. Therefore, significant attention should be attached to the modification strategy of natural minerals, which is of great importance not only to the field of environmental remediation, but also to the industrialization application of catalysts, and the comprehensive utilization level of mineral resources.

(2) In practical applications, it is considered that auxiliary means are used to enhance the properties of natural mineral materials. For instance, in terms of photocatalytic sterilization of natural mineral materials, there may exist some problems, such as the gradual evolution of bacterial resistance to ROS, the toxicity of high concentration of ROS to normal tissues and cells, the biological safety of photocatalytic materials, and the poor penetration ability of visible light or even near-infrared light to deep tissue infection. Therefore, it can be combined with photothermal therapy, microwave-assisted therapy, ultrasound therapy, and other strategies to improve the property.

(3) Natural mineral materials provide new solutions for enhanced environmental remediation. However, no matter pristine mineral materials or modified natural mineral materials, the complex characteristics of its components make obstacles to the practical application in the field of environmental remediation. In future developments, it is necessary to study the active components, such as using density functional theory calculations to determine the active surface.

(4) Natural mineral catalysts can significantly improve the ability of Fenton catalysis and persulfate oxidation to remove pollutants, but in practical applications, appropriate mineral materials should be selected according to the property of pollutants.

(5) The advantage of natural minerals is that they are conducive to reuse after recovery. However, recycling experiments have found that the catalytic activity of natural minerals decreases after limited realization. Therefore, it is necessary to conduct in-depth research on the active components of natural minerals to prevent the loss of active components of natural mineral catalysts and improve the reuse ability.

**Author Contributions:** Conceptualization, K.L.; formal analysis, N.K., W.Z., Z.Q., Y.L., Z.W. and Q.L.; writing—original draft preparation, N.K.; writing—review and editing, Y.L. and K.L.; supervision, project administration and funding acquisition, K.L. All authors have read and agreed to the published version of the manuscript.

**Funding:** This work was supported by the National Natural Science Foundation of China (51672312, 5180808 and 21972171), the Fundamental Research Funds for the Central Universities, South-Central Minzu University (CZP22001), the Research Project of Chongqing Education Commission Foundation (KJQN201800826), Science and Technology Research Program of Chongqing Municipal Education Commission of China (KJZD-K202100801).

**Institutional Review Board Statement:** Not applicable.

**Informed Consent Statement:** Not applicable.

**Data Availability Statement:** Not applicable.

**Conflicts of Interest:** The authors declare no conflict of interest.

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
