# Peer review of "Recent Progress of Natural Mineral Materials in Environmental Remediation"

_catalysts, doi:10.3390/catal12090996_

Round 1

Reviewer 1 Report

The topic of this short review (as Natural Minerals for Environmental Remediation) is very interesting. In the literature, this way of results presentation is rare. The manuscript reviews a sufficient number of investigations (68 Ref.). But unfortunately, the structure of the review is not perfect. There are also other disadvantages. A major revision is required.

Major remarks:

1 It is difficult to perceive the information due to the lack of abbreviations in most cases. I strongly recommend that the authors, when mentioning the name of minerals, indicate the chemical composition of them. The readers of the Catalysts journal are mostly chemists, not geologists.

2 It is very difficult to view the contents of the figures, the captions in some fragments are not visible. Some information (for example, the EPR data in Fig.3C and etc.) is redundant for this review and is not discussed in the text. Then why is it shown? The authors note that they wrote a reference review presenting the main ideas. The pictures must match the level of Graphical Abstracts.

3 The attention of the authors is occupied by the three main directions in the use of natural materials. These are various photochemical processes (1), Fenton processes (2) and adsorption processes (3). These processes are reviewed in Chapter 3, and re-described in Chapter 4. When reading this text, there is a feeling of the repetitions of information but only in a section with a different title.

4 I advise to remove the Section 3, and divide the information according to the processes under study (1)-(3). I would like to draw the authors' attention to the fact that in the case of photocatalysts consisting of two semiconductors, it is rather difficult to answer the question of what is the support and what is the catalyst (for example, BiOCl or TiO2 on Fig.2B and etc.). In addition, the solids used in Fenton reactions are traditionally  referred to as Fenton catalysts (not as activators), although they play an activating role.

5 For example, the Section 3.1 might be devoted to photocatalytic processes, with subheadings describing the applications of photocatalytic processes (sterilization, degradation and etc.). the Section 3.2 might be devoted to the Fenton catalysts and etc.

6 The type of chemical (or adsorbing) process should be pointed out in the Table 1 on the strategy of combination of minerals with other synthetic materials.

7 The text contains a number of inaccuracies and typos, which distorts the content of the written phrases. The text needs to be carefully checked.

8 I recommend correcting the Abstract and Conclusions in accordance with the text changes made.

Reviewer 2 Report

In the paper titled “Recent Progress of Natural Mineral Materials in Environmental Remediation” the authors present the role of the natural mineral materials for the degradation of organic contaminants, volatile organic compounds and heavy metals removal. The review illustrate a classification of natural mineral materials, the role of these in the field of environmental remediation.

I think that this mini-review can be published after the following revisions:

-          The authors introduce the role of semiconductor materials as photocatalysts for environmental remediation, they should add some explicative and recent examples adding references.

-          The review is poor of references. Only 68 references have been cited. Bibliography must be revised

-          The discussion of the different classes of natural mineral materials must be improved providing also examples of preparation methods and experimental use of them

-          An introduction of the photocatalytic process must help people not very confident on the field to understand better the process and the role of the materials

Round 2

Reviewer 1 Report

The paper can be published, the authors have made the required corrections to the text. Misprints can be corrected at the next publication stage.